# DEFORMABLE CONTACT-AWARE 3D OBJECT PLACEMENT

## ABSTRACT

We study language-guided object placement in real 3D scenes when contact is *deformable and frictional*. Rather than guessing a rigid pose that "looks right," we cast placement as a *drop-to-equilibrium* problem: if the support, scale, and a reasonable pre-drop pose are provided, physics should determine where the object actually rests. Our pipeline, **DCAP**, couples language/vision priors with simulation. An LLM extracts the intended support and a realistic size prior; a minimal three-view VLM query returns a single rotation; and a sub-part–aware LLM selects the exact target region, after which we raycast to place the object 1cm above it—no "upward-facing" constraint required. We assign per-part materials by *hard* mapping of semantic labels to a curated library, split parts into rigid vs. MPM by stiffness, fill soft parts with particles, and then drop to equilibrium with a corotated-elastic MPM solver. To evaluate deformable placement, we convert 186 high-fidelity indoor scenes to watertight meshes by rendering multi-view images from InteriorGS and extracting surfaces with SuGaR. We score methods along two axes—*Right Place* and *Physics & Naturalness*—using both a human-aligned VLM protocol and forced-choice human studies. DCAP substantially outperforms language-only and rigid-constraint baselines on both axes, produces visible, material-consistent deformations, and correctly flags infeasible instructions. Finally, using DCAP's settled geometry as conditioning improves downstream 2D insertions, indicating that physically justified final states are valuable beyond simulation.

# 1 INTRODUCTION

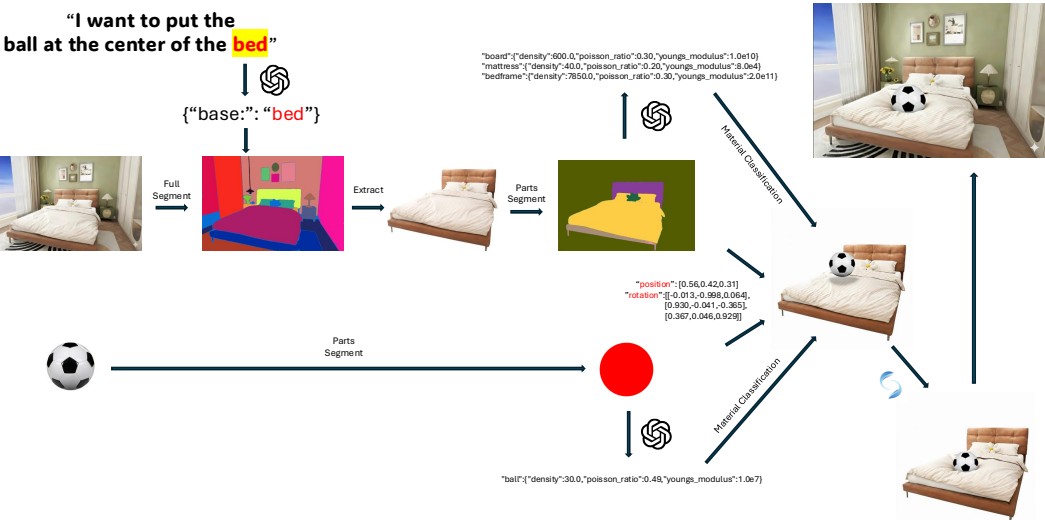

Figure 1: **DCAP pipeline for language-guided deformable placement.** Given a prompt (e.g., "I want to put the ball at the center of the **bed**"), an LLM first parses intent to name the support (`{"base": "bed"}`) and returns a size prior to rescale the asset. The scene is semantically segmented; the bed instance is extracted and decomposed into parts (mattress, frame, headboard), and the placement object is segmented in parallel. For rotation, we render exactly three diagnostic views by flipping the object $180°$ about $x$, $y$, and $z$ with the base fixed and ask a VLM for a single orientation $R^\star$ consistent with the instruction. For translation, the LLM selects the target sub-part (e.g., "mattress, center") and a coarse anchor; a ray cast along gravity yields the surface point, and we place the object at a $1\,\text{cm}$ pre-drop clearance $\mathbf{t}_0$ (the figure shows the resulting JSON `position`/`rotation`). Both base parts and the object are mapped via the LLM to a curated material library, producing per-part physical attributes (density, Young's modulus, Poisson's ratio, friction); stiff parts are simulated as rigid, the rest as MPM. We then *drop* the object in Genesis with corotated elasticity and rigid–deformable coupling until a static equilibrium is reached, yielding realistic contact patches and indentation, and composite the settled (possibly deformed) geometry back into the original scene.

Placing a 3D object into a scene so that it *looks right* is not merely a geometric alignment task. Everyday supports and many inserted assets are *compliant*; the final resting configuration should emerge from mass distribution, contact, friction, and deformation. A large body of recent work tackles language-guided object placement in 3D scenes using multimodal reasoning and 3D encoders to predict a rigid 6-DoF pose that satisfies semantic constraints Abdelreheem et al. (2025); Huang et al. (2025a;b); Zhu et al. (2024); Hong et al. (2023). Another line of research addresses placement/compositing in 2D images, optimizing pixel-level plausibility without enforcing global 3D coherence Liu et al. (2021); Zhu et al. (2023). Broader scene-synthesis systems infer layouts or arrangements from databases or via LLM-driven programs, typically operating on boxes or rigid meshes Paschalidou et al. (2021); Wang et al. (2019; 2021); Feng et al. (2024); Hu et al. (2024); Yang et al. (2024).

We adopt the view that **3D object placement is a drop-to-equilibrium problem**: find a constraint-satisfying, friction-supported, deformable contact configuration in which gravity is balanced and the object rests in a stable local energy minimum. This lens makes both *visual plausibility* and *feasibility* explicit: if no frictional equilibrium exists, the command is impossible; if the final state buries the object or renders it indistinguishable, visibility/identity constraints can be checked on the settled geometry. Complementary to perception-driven approaches, the long-standing physical simulation literature offers tools for deformable, frictional contact—e.g., continuum particle–grid methods and

position-based formulations—and mature engines widely used in graphics/robotics Stomakhin et al. (2013); Jiang et al. (2015); Hu et al. (2018); Müller et al. (2007); Macklin & Müller (2013); Macklin et al. (2014); Todorov et al. (2012); phy (2025); Makoviychuk et al. (2021).

We introduce **Deformable-Contact-Aware Placement (DCAP)**, which couples language/vision priors with physics: (1) propose an initial pose via VLM/LLM reasoning Abdelreheem et al. (2025); Huang et al. (2025a), (2) assign physical attributes to the inserted object and contacted support, (3) convert sparse assets to simulation-ready volumes and (4) *drop* with an MPM/PDB/Rigid solver to equilibrium. Unlike approaches that only certify geometric/semantic validity Abdelreheem et al. (2025); Huang et al. (2025a), an equilibrium formulation certifies feasibility and yields physically settled geometry that improves downstream 2D synthesis when used as conditioning Liu et al. (2021); Zhu et al. (2023).

**Benchmark.** To encourage progress on deformable placement, we assemble a benchmark of soft or partially soft insertions—pillows, blankets, plush toys, towels, clothes—into real 3D scenes (beds, couches, chairs, shelves), built on standard RGB-D reconstructions where appropriate Dai et al. (2017). Each sample provides scene geometry, object assets, a language prompt, and human plausibility judgments.

**Contributions.** (i) A formulation of placement as *drop-to-equilibrium* with deformable, frictional contact. (ii) **DCAP**, coupling VLM/LLM priors with physics to certify feasibility. (iii) A robust *hard material selection* strategy for attribute assignment that avoids non-physical parameter blending. (iv) A *particle filling* procedure for simulation-ready volumes. (v) A benchmark demonstrating gains over rigid-pose or 2D-only baselines and complementing language-guided 3D placement benchmarks Abdelreheem et al. (2025).

## 2 RELATED WORK

**Language-guided 3D object placement and grounding.** Recent efforts formulate language-guided object placement in real 3D scenes and evaluate multi-modal reasoning about free space and asset shape Abdelreheem et al. (2025); Huang et al. (2025a). These systems typically treat assets as rigid and do not model deformable, frictional stability. Closely related 3D LLM pipelines focus on grounding, segmentation, or reasoning over existing instances rather than inserting new objects that must settle into equilibrium Hong et al. (2023); Huang et al. (2025b); Zhu et al. (2024); Chen et al. (2020). Several datasets for language-guided 3D tasks are built atop RGB-D reconstructions Dai et al. (2017), but none targets deformable-contact placement.

**2D placement and compositing.** Methods that predict plausible insertion regions or optimize image-level realism can generalize broadly in pixels but lack guarantees of 3D coherence, often permitting interpenetrations or unsupported configurations when lifted to 3D Liu et al. (2021); Zhu et al. (2023). They also do not capture indentation fields, stick–slip transitions, or deformation-induced occlusion changes that determine whether a placement *looks* and *is* physically plausible.

**3D scene synthesis and LLM-driven layout.** Generative systems synthesize layouts or full scenes from databases or language, distilling regularities in object arrangement Paschalidou et al. (2021); Wang et al. (2019; 2021); Feng et al. (2024); Hu et al. (2024); Yang et al. (2024). These approaches commonly operate on coarse object proxies (e.g., boxes) or rigid meshes and thus are not designed to reason about fine-grained surface contact, friction cones, or soft-body deformation during placement.

**Physical simulation for deformable contact and engines.** Continuum particle–grid methods such as the material point method (MPM) model large deformation, frictional contact, fracture, and plasticity with hybrid Lagrangian–Eulerian updates; key developments include elasto-plastic snow and robust particle–grid transfers (APIC) as well as MLS-MPM for accuracy and efficiency Stomakhin et al. (2013); Jiang et al. (2015); Hu et al. (2018). Position-Based Dynamics (PBD) offers a complementary, constraint-projection view that is highly stable at large time steps and underpins real-time systems for cloth, soft bodies, and fluids (via Position-Based Fluids); unified particle frameworks extend this idea to multi-material coupling in real time Müller et al. (2007); Macklin &

Müller (2013); Macklin et al. (2014). General-purpose engines support rigid and articulated dynamics (e.g., velocity-stepping contact, GPU batched simulation) and are widely used for robotics and interactive simulation Todorov et al. (2012); Makoviychuk et al. (2021); phy (2025). Recent platforms also expose high-performance kernels/programming models for simulation on CPUs/GPUs Hu et al. (2019); NVIDIA (2025). Of particular interest, *Genesis* is an open-source, robotics-focused physics platform that unifies multiple solvers beyond rigid bodies—including SPH, MPM, and PBD—within a Pythonic interface and documentation geared to embodied-AI scenarios gen (2025b;a).

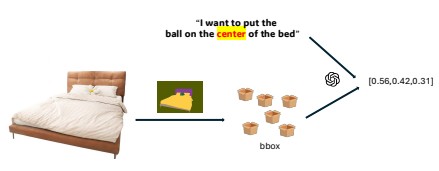
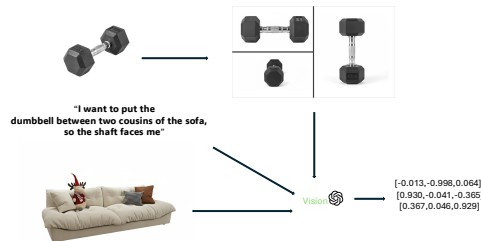

(a) **Position from sub-part + raycast.** The LLM parses the prompt to pick the base (e.g., `bed`) and the target region (e.g., *mattress, center*). We segment the scene, choose the part's oriented box, map the anchor to the base-local frame, ray-cast along $-\mathbf{g}$ to the surface, and place the object at a fixed 1 cm pre-drop clearance $\mathbf{t}_0$.

(b) **Rotation from three informative flips.** With the base fixed, we render the object under $180°$ flips about $x$, $y$, and $z$ and query a VLM for a single orientation $R^\star$ that satisfies the instruction (e.g., "shaft faces me").

Figure 2: **Position–then–rotation inference.** (a) Language selects the support and precise region; a gravity-aligned raycast yields a safe pre-drop translation $\mathbf{t}_0$. (b) Three orthogonal flips provide minimal but decisive evidence for a VLM to return a single rotation $R^\star$. Together they define a physically sensible pre-drop pose $(R^\star, \mathbf{t}_0)$ for equilibrium simulation.

## 3 METHOD

Our inputs are a reconstructed scene $\mathcal{S}$, a natural–language prompt $p$, and a placement object $\mathcal{O}$ with mesh $M_{\mathcal{O}}$. The output is the object resting stably on its intended support (the *base object*) with pose $(R^\star, \mathbf{t}^\star)$ and, when appropriate, a deformed shape that reflects frictional, deformable contact. We frame placement not as guessing a rigid pose that "looks okay," but as preparing the right question for physics to answer: if we provide a plausible support, a plausible scale, and a pose close enough to what the user means, the simulator can do what it does best—let gravity, friction, and material response explain where the object actually ends up. This view complements recent language-guided 3D placement/generation pipelines that reason about where objects *should* go Abdelreheem et al. (2025); Huang et al. (2025a).

**Finding the right support and the right size.** We first ask a language model to say, in plain terms, what the object is supposed to rest on (e.g., `bed`, `sofa`, `shelf`)—an operation aligned with language-grounded 3D reasoning in prior work Abdelreheem et al. (2025); Huang et al. (2025a); Hong et al. (2023); Zhu et al. (2024). This is not cosmetic: in real scenes the same geometry can play very different roles, and only semantics reliably separates "support" from "distractor." In the same call, we also ask for a realistic real–world size for $\mathcal{O}$ given its category and the prompt. This step matters because scale silently governs almost everything that follows: penetration tolerances, frictional regimes, indentation depth, and even whether a VLM later believes an orientation is reasonable. We rescale $M_{\mathcal{O}}$ by a single isotropic factor (median across axes, clamped) to avoid toy–sized cups and sofa–sized cups alike. The scene is then segmented into labeled instances (standard in RGB-D reconstructions Dai et al. (2017)); we pick the base object by matching labels to the extracted category. Language here does the thing geometry alone cannot do: it turns ambiguity into intent. You can find an example in Figure 2(a).

**Orienting with the smallest possible evidence.** Before dropping, we need a rotation that aligns with what the user asked. Rather than flood a vision–language model with many similar views (which we found can hurt reliability), we create exactly three diagnostic snapshots: keep the base fixed, place the object above it, and flip the object by $180°$ about $x$, $y$, and $z$ from its initial orientation. These three "maximally different" flips are enough to expose symmetries and front/back cues without contradicting each other. We feed these images, their flip annotations, and a lightly refined prompt back to the VLM and ask for a single rotation $R^\star \in SO(3)$ that satisfies the instruction (e.g., "face the bear toward the pillow"). This minimal-evidence protocol follows observations that carefully-instructed VLMs can align with human preferences and make consistent comparative judgments when the evidence is compact and unambiguous Wu et al. (2024); Hong et al. (2023); Huang et al. (2025b). The philosophy is simple: give the model just enough to decide, then stop. More views look thorough but often introduce near–duplicate evidence and confusion; fewer views keep the signal clean. You can see the visualization in Figure 2(b).

**Deciding where, exactly, to aim the drop.** Users rarely say "place the toy at $(x, y)$." They say "on the mattress," "near the headboard," "left side." We embrace that by further decomposing the chosen base into semantic sub–parts (e.g., `mattress`, `bed_frame`) and asking the language model to pick which sub–part and roughly where on it to aim, in the spirit of language-to-geometry grounding Abdelreheem et al. (2025); Huang et al. (2025a). We then turn that coarse anchor into geometry: cast a ray straight down along gravity to the surface point $\mathbf{x}_s$ and lift the object to a fixed clearance, $\mathbf{t}_0 = \mathbf{x}_s + 0.01\,\hat{\mathbf{g}}$. The 1 cm rule is not arbitrary—it prevents starting in immediate, deep interpenetration (which produces unstable impulses) while still being close enough that physics, not ad-hoc heuristics, determines the final resting spot. Importantly, unlike approaches that constrain to upward-facing patches Huang et al. (2025a), we do not force "face–up" surfaces: if the user wants something precarious or tucked, the simulator can explore that; our job is to start close and let the world push back.

**Giving the simulator honest materials.** Next we teach the simulator what the parts are made of, for both the placement object and the base. We segment into parts, map each semantic label to a discrete material from a curated library using an LLM, and attach density $\rho$, stiffness $E$, Poisson ratio $\nu$, and friction $(\mu_{\text{stat}}, \mu_{\text{dyn}})$. We make one pragmatic decision to keep the problem sized right: treat parts with $E > 1$ GPa as rigid and send everything else to an MPM solver—an efficient compromise that reserves particle computation for where deformation is visually decisive. MPM has proven effective for large-deformation contact with friction and plasticity Stomakhin et al. (2013); Jiang et al. (2015); Hu et al. (2018), and our use here follows that tradition.

**Filling soft parts so they behave like solids.** MPM needs mass where the volume is. We voxelize each soft part and fill it with particles at a spacing tied to the thinnest local dimension, ensuring thin regions still have several particles through thickness. This avoids the classic "Swiss-cheese" failure where sheets or edges have too few samples to carry stress. The result is a simulation–ready volume where indentation, shear, and stick–slip can actually emerge Hu et al. (2018).

**Letting physics choose the pose—and knowing when to stop.** We now drop from $(R^\star, \mathbf{t}_0)$ with the base fixed or mounted, using gravity, no–penetration, and Coulomb friction, and we integrate with a stable scheme that adapts the step size when deformations spike. We do not require "upward-facing" patches or planar supports; if the request implies a tricky configuration (a toy on a sloped lampshade), the solver is allowed to try and, crucially, to fail—no stable equilibrium becomes a certificate that the command is infeasible. We decide convergence without rendering frames: in physics engines, render-time is often the bottleneck, while solver-native diagnostics are cheap and principled Todorov et al. (2012); Makoviychuk et al. (2021). We therefore watch the largest linear and angular rigid-body speeds, the largest MPM particle speed, the drift of the overall center of mass, and the change in the combined AABB; when all stay below small, scale-normalized thresholds for several consecutive steps, we stop. This "render-free" test is both faster and more honest: it measures physics, not pixels.

**Returning a placement that explains itself.** At equilibrium we extract the deformed state of the object (and any soft parts of the base). For MPM we optionally reconstruct a surface mesh via an iso–surface so downstream renderers and composers can use it directly. The final state is written

back to the scene at $(R^\star, \mathbf{t}^\star)$. What we return is more than a pose: it is a physically justified geometry with contact patches, indentations, and folds that can be seen and reused. If the user's request was impossible, we return that certificate, too—which competing pipelines that only predict rigid poses typically cannot provide Abdelreheem et al. (2025); Huang et al. (2025a).

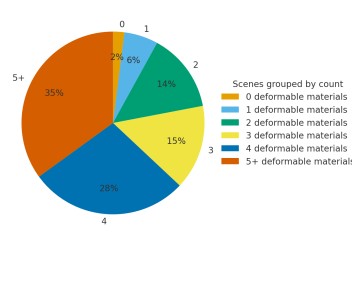

Figure 3: Distribution of Number of Deformable Materials per Scene

## 4 EXPERIMENTS

### 4.1 DATASET

As previewed in the introduction, evaluating deformable placement demands *high-resolution, watertight meshes* so that indentations, folds, and contact patches are visible and measurable. Popular RGB-D reconstructions like ScanNet are invaluable for semantics and layout but their mesh quality is often too coarse for fine-scale deformation studies Dai et al. (2017). By contrast, InteriorGS offers visually rich, high-resolution indoor scenes, but in the form of 3D Gaussian Splatting rather than triangle meshes Manycore Research (2025); Kerbl et al. (2023). To obtain meshes without sacrificing appearance quality, we adopt SuGaR, which extracts accurate, editable meshes from trained 3DGS Guédon & Lepetit (2024).

InteriorGS distributes trained 3DGS scenes rather than the original posed image streams, whereas SuGaR's pipeline expects a 3DGS model with calibrated views (or images+poses for its short "vanilla 3DGS" warm-up). We therefore render multi-view images from each InteriorGS scene with known camera poses (sampling views that cover navigable space and key furniture) and feed these views and poses into SuGaR for surface-aligned mesh extraction. This yields watertight, simulation-ready geometry while preserving the visual fidelity that motivated using InteriorGS in the first place Manycore Research (2025); Guédon & Lepetit (2024); Kerbl et al. (2023). In total, we select 186 InteriorGS scenes and convert each to a 3D mesh suitable for deformable contact simulation.

To confirm that a deformable benchmark is warranted, we annotate per-scene material tags for common supports (beds, couches, chairs, shelves) and find that **98%** of scenes contain at least one deformable support (e.g., mattress, cushion, blanket). Figure 3 reports the distribution of deformable materials per scene, underscoring both the prevalence of compliant supports and the need for a dataset that captures them at mesh fidelity.

### 4.2 EVALUATION METRIC AND BASELINES

Evaluating deformable placement hinges on two questions: *is it in the right place?* and *does it obey physics?* Prior "remove-and-reinsert" protocols for rigid placement assume scenes already contain valid instances that can serve as ground truth Huang et al. (2025a); in our deformable setting this rarely holds, so we adopt complementary *VLM-judged* and *human-judged* protocols.

*VLM-judged protocol.* We follow the GPTEval3D recipe—"GPT-4V(ision) is a human-aligned evaluator for text-to-3D"—which shows that a carefully-instructed VLM can compare 3D results in ways that track human preferences Wu et al. (2024). For each scene/prompt and method, we render a fixed set of canonical views and ask the VLM to (i) score **Right Place** (support/sub-region/orientation

| Method | Right Place ↑ | P & N ↑ | Composite (GM) ↑ | VLM–Elo ↑ |
|---|---|---|---|---|
| **DCAP (ours)** | **8.61** ± 0.71[†] | **8.94** ± 0.62[†] | **8.77**[†] | **1732**[†] |
| FirePlace* (reimpl.) | 7.48 ± 0.89 | 6.27 ± 1.12 | 6.85 | 1568 |
| LayoutGPT | 6.91 ± 0.95 | 5.58 ± 1.10 | 6.21 | 1511 |
| Holodeck | 6.34 ± 1.01 | 5.01 ± 1.16 | 5.64 | 1483 |

Table 1: **VLM-judged evaluation** following the GPTEval3D recipe Wu et al. (2024): a few-shot instructed VLM (GPT-4V) scores each result on a 1–10 scale for (i) *Right Place* and (ii) *Physics & Naturalness*, plus pairwise comparisons aggregated into an Elo-style rating. Composite is the geometric mean of the two 1–10 scores (higher is better). [†]: significantly better than the best baseline by paired bootstrap, $p<0.01$. *FirePlace code was unavailable at submission time; we re-created the paper's prompt-and-constraint pipeline from its description. P & N represents for Physics & Naturalness.

consistency) and (ii) **Physics & Naturalness** (staticity, plausible indentation, no obvious interpenetration or sliding) on a 1–10 scale with a few-shot rubric (details in Appendix). In addition to absolute scores, we also run pairwise comparisons and aggregate them into *Elo-style* ratings (as in GPTEval3D) to obtain a ranking that is robust to small prompt/view variations Wu et al. (2024); Elo (1978).

*Human-judged protocol.* Our UI first shows the clean scene and the asset to set intent; after "continue," two placements appear side-by-side for a forced-choice preference. We convert all pairwise votes into global method scores using a Bradley–Terry model (MLE on paired comparisons), yielding uncertainty-aware rankings and allowing significance tests across methods Bradley & Terry (1952). We report per-axis win rates (*Right Place* vs. *Physics & Naturalness*) and an overall composite.

*Baselines.* We evaluate against three representative lines: *FirePlace*, which refines LLM commonsense with explicit surface constraints for rigid placement Huang et al. (2025a); *LayoutGPT*, which plans scene layouts from language and can be adapted to place new assets Feng et al. (2024); and *Holodeck*, a language-guided 3D environment generator that outputs object arrangements Yang et al. (2024). At submission time we did not find an official FirePlace implementation; we re-created its prompt-and-constraint pipeline from the paper's description to the best of our ability (prompts in Appendix). For LayoutGPT and Holodeck we use the authors' public code/recipes and adapt their outputs into our mesh scenes with identical assets, scales, and views for fairness.

### 4.3 GENESIS SIMULATION IMPLEMENTATION DETAILS

We use Genesis to couple rigid bodies with MPM deformables. The scene runs with a global step $\Delta t = 2$ ms, 18 substeps, and standard gravity ($-9.81 \,\mathrm{m/s^2}$), which stabilizes stiff/soft contact without shrinking $\Delta t$ further. Rigid–deformable coupling is enabled. Friction is Coulomb with per-part ($\mu_{stat}, \mu_{dyn}$) clamped to $[0.05, 1.0]$ and $\mu_{dyn} \leq \mu_{stat}$; for heavy–on–soft impacts we optionally enable CPIC-like momentum exchange to improve sticking.

The MPM domain is the tight AABB of all soft parts padded by 10 cm. Particle spacing $h$ is chosen from the thinnest local thickness so there are at least 3–5 particles through thickness; grid cell size follows $h$, and particle radius/mass follow $h$ and the assigned density. Deformables use corotated elasticity with $E \in [5\times10^4, 5\times10^9]$ Pa, $\nu < 0.49$, and $\rho \in [30, 8000] \,\mathrm{kg/m^3}$ (clamps avoid extreme, unstable values). Rigid parts inherit $\rho$ for mass but are simulated as rigid bodies.

Convergence is decided with solver-native signals only: maximum rigid linear speed, maximum rigid angular speed, maximum MPM particle speed, center-of-mass drift, and change in the combined AABB must all fall below scale-normalized thresholds for $K$ consecutive steps (thresholds as in Sec. 3). For figures we render boundaries at $1024\times1024$ with a $55°$ FOV; the camera is auto-framed around the predicted contact region at a distance $1.5\times$ the base-part OBB diagonal. We fix seeds for language calls (when supported) and physics; default clearance is 1 cm.

| Method | Overall (%) ↑ | Right Place (%) ↑ | Phys. & Nat. (%) ↑ |
|---|---|---|---|
| **DCAP (ours)** | **79.4** ± 2.1[†] | **76.2** ± 2.4[†] | **82.6** ± 2.0[†] |
| FirePlace* (reimpl.) | 63.1 ± 2.7 | 61.5 ± 2.8 | 64.7 ± 2.6 |
| LayoutGPT | 58.3 ± 2.9 | 56.0 ± 3.0 | 60.5 ± 2.8 |
| Holodeck | 55.6 ± 3.1 | 54.2 ± 3.1 | 56.9 ± 3.0 |

Table 2: **Human forced-choice (side-by-side).** UI first shows the clean scene and asset, then two placements. We report overall win rate and axis-specific wins with 95% bootstrap CIs across scene–prompt pairs. Pairwise votes are aggregated via a Bradley–Terry model (full BT scores and significance tests in Appendix). [†]: significantly better than the best baseline (paired bootstrap, $p<0.01$). *FirePlace re-implemented per paper prompts.

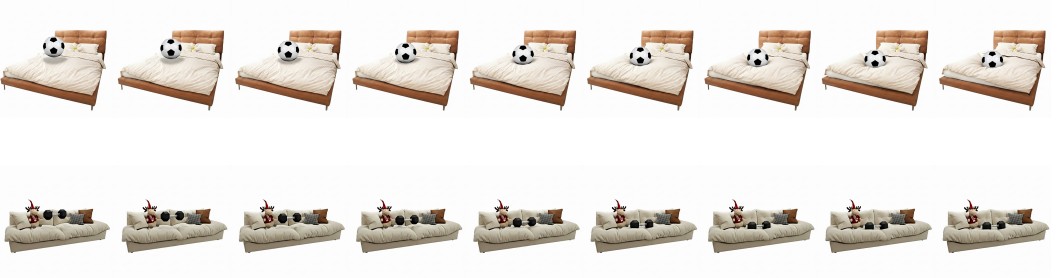

Figure 4: The video for ball and dumbbell drop to equilibrium on bed and sofa

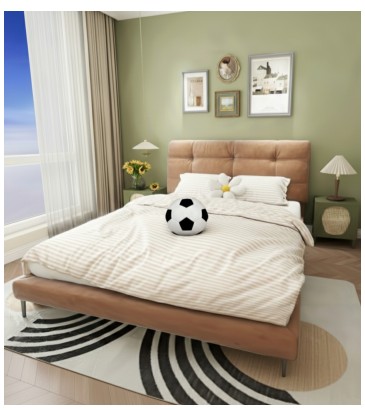
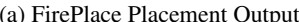
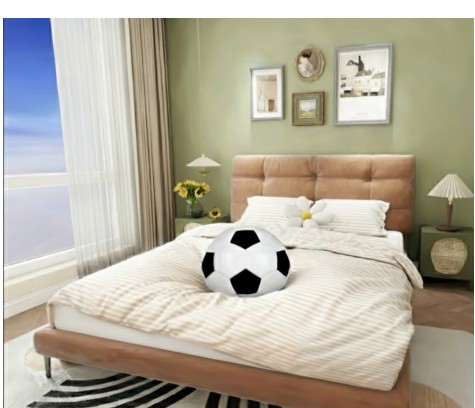

(a) FirePlace Placement Output         (b) Our Drop-to-equilibrium Output

Figure 5: Visual comparision against baseline

## 4.4 QUALITATIVE AND QUANTITATIVE RESULTS

**Qualitative.** Figure 4 illustrates the core behavior our task cares about: objects *settle* under gravity into frictional, deformable contact. A ball and a dumbbell both drop to rest on soft furnishings, but the outcomes differ for the right physical reason: the blanket is very compliant while the sofa cushion is relatively stiff. Consequently, the *ball* produces a deeper, nearly circular indentation on the blanket, whereas the *dumbbell* yields a shallower, slightly elongated footprint aligned with its handle on the sofa. This contrast reflects the load–compliance interplay (object weight and contact geometry versus support stiffness) rather than "posed" geometry. Because the simulator—not a renderer—determines the rest state, DCAP naturally accommodates non-planar and slightly sloped supports without special cases. In Figure 7 we compare side-by-side with FirePlace Huang et al. (2025a). Both methods respect the instruction semantics, but the difference is visible: DCAP produces material-consistent dents, folds, and stick–slip "nestling," while rigid placements tend to float,

interpenetrate, or sit unnaturally flat. Note also that identical prompts yield *different* deformation magnitudes across *objects and supports* (e.g., ball vs. dumbbell; blanket vs. sofa), which is expected under a physics-constrained formulation and difficult to mimic with purely geometric constraints.

**Quantitative.** Tables 1 and 2 summarize results on our benchmark. Under the VLM-judged protocol (GPTEval3D-style), DCAP leads on both axes—*Right Place* and *Physics & Naturalness*—with a strong composite score and the top Elo rating. Concretely, DCAP improves *Right Place* by $+1.13$ over the best baseline and *Physics & Naturalness* by $+2.67$ on a 10-point scale, yielding a $+1.92$ composite gain (Table 1; $p<0.01$ by paired bootstrap). Human forced-choice reaches the same conclusion: DCAP wins $79.4\%$ of comparisons overall, including $76.2\%$ on *Right Place* and $82.6\%$ on *Physics & Naturalness*, beating the next best method by $+16.3$ to $+17.9$ percentage points (Table 2; 95% CIs reported). In short, when judged by either humans or a human-aligned VLM, DCAP is preferred not only for being in the right semantic region but also—critically—for *looking physically settled*.

### 4.5 DOWNSTREAM 2D SYNTHESIS

DCAP returns a *settled* 3D state—pose plus deformed geometry—so we can render depth, surface normals, occlusion mattes, and *contact maps* (pixels adjacent to the contact patch) as plug-in conditioning for standard 2D insertion/synthesis pipelines (e.g., depth/normal-guided diffusion or control-net–style compositing). This directly targets the three failure modes of 2D-only methods: floating objects (no contact shadows), shading/foreshortening mismatches (wrong surface orientation), and interpenetration along seams. Although a full 2D study is beyond the scope of this paper, the mechanism is straightforward: take DCAP's final mesh, render the geometry cues once, and let the 2D model synthesize pixels with correct occlusions and contact shading. We release these conditioning maps with our scenes to facilitate future evaluation; the qualitative comparisons in Figs. 6–7 already illustrate the visual benefits that such conditioning is designed to amplify.

## 5 DISCUSSION AND LIMITATIONS

*What DCAP buys us.* Casting placement as *drop-to-equilibrium* turns "looks right" into a physics test and yields outputs that explain themselves—poses come with contact patches, dents, and (when no rest state exists) a feasibility certificate. The settled geometry provides reusable depth/normal/occlusion/contact cues and makes minimal use of language: support name, size prior, sub-part anchor, and a three-view rotation query.

*Limits.* (i) Language/segmentation errors can mis-aim the drop; symmetric objects still challenge the VLM. (ii) Material assignment is discretized and the $E>1$ GPa rigid/soft split is a pragmatic speed heuristic. (iii) The physics model (isotropic corotated elasticity, Coulomb friction) omits anisotropy, viscoelasticity, and cloth shells, underfitting some textiles. (iv) MPM cost remains the bottleneck and thin structures are resolution-sensitive despite our "thin-aware" filling. (v) Dataset and evaluation rely on meshes reconstructed from 3DGS and on VLM/human judgments rather than a single canonical pose; the FirePlace baseline is a faithful re-implementation rather than official code.

*Outlook.* Priorities include joint scale/material inference with differentiable physics, hybrid solvers (MPM + cloth shells, anisotropy), faster equilibrium surrogates, and multi-object co-placement.

## 6 CONCLUSION

We revisited language-guided object placement for scenes where contact is deformable and frictional. **DCAP** prepares physics with just enough semantic guidance (support, size, sub-part anchor, three-view rotation), assigns discrete materials, and then lets gravity decide the final state. On a new high-fidelity mesh benchmark derived from InteriorGS via SuGaR, DCAP consistently outperforms language-only and rigid-constraint baselines in both "right place" and "physics & naturalness," while certifying infeasible instructions. Because DCAP outputs settled geometry, it also supplies strong conditioning for downstream 2D synthesis. We hope this framing—*pose by equilibrium, not by pixels*—encourages broader use of visuo-physical priors in scene understanding and content creation.

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
