# OpenReview forum: "Deformable Contact-Aware 3D Object Placement"
_ICLR.cc/2026/Conference — Submitted to ICLR 2026_

### Official Review · Reviewer_57bo · 2025-10-29

**Soundness:** 2
**Presentation:** 1
**Contribution:** 2
**Rating:** 2
**Confidence:** 5

**Summary:**

This paper proposes a method for arranging deformable objects using large language models (LLMs). Previous similar studies only considered rigid bounding boxes without accounting for contact-induced deformation. This paper addresses this issue by proposing a collaborative solution involving multiple components. However, the overall writing, formatting, diagramming, and font sizes in the figures severely hinder readability. The provided qualitative experimental demos are also too sparse to reliably assess their generalization capabilities. Furthermore, there is a lack of evidence demonstrating whether the employed LLM can effectively meet the requirements.

**Strengths:**

The research question is novel and meaningful.

**Weaknesses:**

1. The paper writing is too poor. Both the writing and diagrams include significant issues. For example, the font size in the images is too small. These problems make the paper hard to read.
2. The provided qualitative experimental demos are also too sparse to reliably assess their generalization capabilities. The authors need to provide more evidence to prove the robustness of such a complex system.
3. There is a lack of evidence demonstrating whether the employed LLM can effectively meet the requirements. The authors need to provide a more complete evaluation of the LLM to prove that the selected LLM has a strong capability to understand the setting.

**Questions:**

See weakness.

---

### Official Review · Reviewer_k4op · 2025-10-31

**Soundness:** 2
**Presentation:** 2
**Contribution:** 2
**Rating:** 4
**Confidence:** 2

**Summary:**

Uses segmentation and VLM queries to try and solve object placement in the problem of 2D image editing, trying to simulate properties of the different objects to give good contacts and realistic deformations.

**Strengths:**

Lots of description on what is being done helps the readers through a difficult method improving the clarity, but this significant usage of page space limits your ability to demonstrate the significance of the method.

**Weaknesses:**

Lack of qualitative evaluations for a visual task, makes it hard to judge this work and understand the improvement over prior works. Even some in the supmat would be sufficient. The results shown so far look interesting and I would like to see more. Any more diagrams that can help more quickly convey the method so that you can spend more time demonstrating your method's results would also be beneficial.

**Questions:**

Are there more evaluations that can be run aside from human preference and VLM evals? This is a task I'm not as familiar with but I'd love to see more quantitative evals if any more benchmarks exist.

---

### Official Review · Reviewer_YM5r · 2025-11-01

**Soundness:** 2
**Presentation:** 1
**Contribution:** 2
**Rating:** 2
**Confidence:** 3

**Summary:**

The authors introduce an approach to place objects into simulated 3D scenes using language prompts specifying the desired position of the object. From the prompt, the approach makes use of LLMs and VLMs to visually locate the intended support of the object in a rendered image of the scene, as well as segment the object and its support into parts and infer their physical parameters (e.g. density, Young's modulus) to condition the simulator. The approach further uses LLMs to determine the initial position and rotation of the object for a drop into the scene, then rolls out the simulation until a set of convergence criteria are satisfied. The results are presented to a VLM and a set of human study participants for evaluation, outperforming all of the 3 considered baselines on the "right placement" and "physics & naturalness" metrics.

**Strengths:**

The formulation of placement as "drop-to-equilibrium" is a useful idea for applications such as animation or interior design.
The considered baselines are deemed to be outperformed by human judges.
It is commendable that the authors reimplemented the FirePlace baseline for an additional comparison with a method whose code is not publicly accessible.
The work includes an honest discussion of its limitations.

**Weaknesses:**

The presentation of the work can be improved in several points: improving the citation style, spending more effort on visually appealing figures. Several citations have been formatted incorrectly.
Insufficient examples of the output are provided to allow a sensible assessment of the method's output and performance against baselines. The work could have greatly benefited from the inclusion of supplementary materials in form of videos with qualitative examples of baselines' vs. the proposed method's results, to allow reviewers to form a more informed opinion about its efficacy.
The work cites its appendix in several places, yet no appendix has been uploaded.
The abstract is too detailed. The textual abstract still contains LaTeX code.

**Questions:**

What is the size of the human study in Table 2?
In line 226, you mention that "More views look thorough but often introduce near–duplicate evidence and confusion". It seems counterintuitive to me that providing more evidence to the VLM will produce worse results given a reasonable aggregation scheme such as a majority vote. Do you have more evidence to support this claim?
What is the quality of the LLM semantic label-to-material mapping in the "Giving the simulator honest materials" step?
Which segmentation model is used to segment the image into labeled object instances? How often does this step fail, e.g. due to failure of detection of the object by the segmentation method?

---

### Official Review · Reviewer_Mv2Z · 2025-11-01

**Soundness:** 3
**Presentation:** 2
**Contribution:** 3
**Rating:** 4
**Confidence:** 3

**Summary:**

This paper proposes DCAP pipeline which combines vision/language priors with simulation. The main contributions of the work are as follows:

-   A novel problem formulation which formulates placement as drop-to-equilibrium, adhering to the physical properties of the materials.
-   DCAP pipeline that couples LLM/VLM with physical simulators.
-   A new benchmark that converts 186 high-fidelity InteriorGS scenes into watertight, simulation-ready meshes using SuGaR.

**Strengths:**

-   [S1] **Novel Problem Formulation:** The primary strength of the proposed work is the novel formulation of object placement as a physics-based equilibrium problem. This formula addresses a clear gap in the existing framework that ignores the physics-awareness of the generation.
-   [S2] **New Benchmark:** The proposed method makes a valuable contribution by proposing a high-fidelity benchmark for this task. The authors reconstruct the scenes from InteriorGS and extracted meshing using SuGaR.
-   [S3] **Novel Pipeline:** In the proposed method, high-level semantic reasoning (intent, size and location) is performed by LLMs/VLMs. They utilize a physics simulator to handle the complex contacts and deformations. The proposed method is highly scalable.

**Weaknesses:**

-   [W1] Total inference time is not reported.
-   [W2] "Filling Soft Parts so They Behave Like Solids": The authors do not provide any supporting figures for this. Further, there is no ablation study for this.
-   [W3] The argument of 1cm is not backed by empirical evidence. What was the reasoning behind choosing this value? This design choice should be thoroughly investigated.

**Questions:**

-   [Q1] Please provide more details on the curated material library (Section 3.3). How many materials are included, and what are the ranges and median values for the key parameters
-   [Q2] Can the following experiment be done with the TanksandTemples dataset? Train 3DGS on the dataset, obtain the mesh, and use the proposed pipeline. Please provide reasoning on why it cannot be done.
-   [Q3] Can the proposed pipeline handle out-of-distribution materials?

---

### Meta-Review · Area_Chair_ykUN · 2026-01-06

**Summary:**

This paper proposes DCAP, a pipeline for language-guided object placement in simulated 3D scenes that integrates vision–language models with physical simulation. The method formulates placement as a drop-to-equilibrium process, enabling physically plausible placement and deformation rather than relying on rigid bounding boxes. Given a language prompt, the system uses LLMs and VLMs to identify the target support surface in rendered images, segment objects and supports, infer physical properties (e.g., density and stiffness), and predict an initial pose for simulation. A physics simulator then rolls out the interaction until convergence. The authors also introduce a new benchmark by converting 186 InteriorGS scenes into watertight, simulation-ready meshes using SuGaR. Experiments, including human studies, show improvements over several baselines in placement correctness and physical realism, though the evaluation is limited in scale and qualitative coverage.

**Reviewer Concerns:**

Reviewers raised the following main concerns:
1. Missing or insufficient analysis: Inference time is not reported, and several design choices (e.g., filling soft parts, the 1 cm threshold) lack empirical justification or ablation studies.
2. Insufficient evaluation: The paper provides too few qualitative results and demos for a visual task, making it difficult to assess performance, generalization, and improvements over baselines. Also, there is limited evidence of robustness, and no thorough evaluation of the LLM’s capability or its suitability for the task.
3. Presentation and clarity issues: Writing quality is poor, figures and diagrams are hard to read (e.g., small fonts), citation formatting is inconsistent, and the abstract contains unnecessary detail and LaTeX artifacts.
4. Missing or incomplete supplementary material: The appendix is referenced but not provided, and the lack of supplementary videos limits the ability to evaluate qualitative results.

**Reviewer Scores:**

This submission received all negative scores (4, 2, 4, 2). The authors did not submit a rebuttal. Most reviewers raised concerns about limited evaluation, as well as presentation and clarity issues. The submission also looks incomplete, as it cites appendices that are missing.

---

### Decision · Program_Chairs · 2026-01-26

Reject